# A Comparison of Three Methods for Estimating Abundances of the Globally Endangered African Grey Parrot

**DOI:** 10.3390/biology15010073

**Published:** 2025-12-31

**Authors:** José L. Tella, Iñigo Palacios-Martínez, Guillermo Blanco, Javier Juste, Pedro Romero-Vidal

**Affiliations:** 1Department of Conservation Biology and Global Change, Estación Biológica de Doñana (EBD-CSIC), 41092 Sevilla, Spain; juste@ebd.csic.es (J.J.); pedro.romero@ebd.csic.es (P.R.-V.); 2Department of Evolutionary Ecology, Museo Nacional de Ciencias Naturales (MNCN-CSIC), 28006 Madrid, Spain; inigo.palacios@mncn.csic.es (I.P.-M.); gblanco@mncn.csic.es (G.B.); 3CIBER de Epidemiología y Salud Pública, CIBERESP, 28220 Madrid, Spain

**Keywords:** census, distance sampling modeling, encounter rates, Equatorial Guinea, line transects, parrot abundances, point counts, *Psittacus erithacus*

## Abstract

Parrots are among the most threatened birds globally, and their abundance and population sizes are often difficult to estimate. Researchers have proposed encounter rates—the number of birds or groups detected per hour of observation during walk transects—as reliable surrogates for estimating the abundance of the globally endangered African grey parrot (*Psittacus erithacus*). In this study, we assessed whether car surveys and point counts, two other widely used methods for estimating bird abundance, yield encounter rates comparable to those obtained from walk transects. Using extensive sampling across Equatorial Guinea, we found that encounter rates varied among regions but did not differ among the three survey methods. These results support the combined use of all three methods to estimate the abundance of this globally endangered species and suggest that this approach may be applicable to many other parrot species.

## 1. Introduction

Accurate estimates of bird abundance and population size are essential for addressing a wide range of questions related to evolution, ecology, anthropogenic impacts, and conservation. Several criteria used by the IUCN Red List to classify species’ conservation status rely on spatial and temporal changes in abundance [1], which have important implications for prioritizing global conservation actions. However, obtaining sufficient and reliable population estimates remains a major challenge, and parrots (order Psittaciformes) represent one of the most striking examples in this regard [2]. On the one hand, parrots constitute one of the most diverse avian groups, with approximately 400 species distributed across tropical and subtropical regions worldwide [3]. On the other hand, parrots also include the highest proportion of globally threatened bird species [4,5], accounting for around 30% of all parrot species [1]. Alarmingly, abundance data are available for only about one quarter of the world’s parrot species, and most of these data originate from a single study area per species [2]. Moreover, threatened species have not received proportionate research attention; instead, they often have less information available concerning their abundance trends than non-threatened species [2].

Several factors hinder the measurement of parrot abundance. Many species have reduced and fragmented distributions [2], which require fieldwork in remote, difficult-to-access, and sometimes conflict-affected regions across much of the world’s tropical, subtropical, and temperate continental and island environments [2,6]. Consequently, current efforts to estimate population sizes and abundance trends for parrots lag far behind what is required for accurate assessment under IUCN criteria [2].

Among the methods available to estimate parrot abundance [7], distance sampling surveys based on point counts or line transects walked on foot have been the most frequently applied [2]. Distance sampling modelling allows researchers to estimate population densities while accounting for reduced detectability with increasing distance from the transect line or count point [8]. Although the efficiency of walk transects and point counts in estimating parrot abundance may vary among species [2,9,10,11], the choice of method does not appear to strongly influence the resulting abundance estimates [2]. Nevertheless, the naturally low densities of many parrot species [2], combined with severe population declines driven by habitat loss, legal and illegal trade, and persecution related to crop damage [4,12,13,14], can severely limit the number of detections available for distance sampling analyses. Small sample sizes often yield large confidence intervals, and in some cases researchers may require up to 2000 sampling points to obtain reliable density estimates [15], rendering surveys logistically unfeasible. Roadside car surveys can partly alleviate this limitation by enabling coverage of very large areas and increasing the likelihood of detecting species that occur at extremely low densities or in spatially aggregated patterns [16]. However, car surveys are not feasible in extensive inaccessible areas that lack even unpaved roads suitable for four-wheel-drive vehicles. Furthermore, distance sampling modelling—whether applied to walk transects, car transects, or point counts—relies on a series of assumptions that are frequently violated in practice [7], potentially leading to biased estimates.

To address these limitations, Marsden et al. [17] proposed a simpler approach for estimating parrot abundance and tested it on two African parrot species, the African grey parrot *Psittacus erithacus* and the Timneh parrot *Psittacus timneh*, which were previously considered a single species [18]. These parrots occupy a wide range of tropical forest habitats, from sea level to 2200 m, and occur across a vast area of approximately 3 million km^2^ in Central and West Africa. They nest in tree cavities and may exhibit highly gregarious behavior while feeding on fruits and seeds or gathering at large communal roosts [18,19,20]. Both species are classified as Endangered on the IUCN Red List following severe population declines and local extinctions caused by forest loss and degradation, and primarily by unsustainable international wildlife trade [19,20]. Although their conservation status clearly requires population monitoring, their low densities across patchily distributed rainforests complicate both the sampling of sufficient sites and the collection of enough detections to enable precise density estimation using distance sampling methods [21]. To overcome these challenges, which are common to many parrot species worldwide [2], Marsden et al. [17] used simple encounter rates—the number of birds or groups detected per hour of observation—derived from casual walks and forest stops, as surrogates for parrot density. By conducting walk transects in 10 study areas across the distribution of grey and Timneh parrots, they obtained a strong correlation (r = 0.95) between encounter rates (groups h^−1^) and densities estimated using distance sampling at the same sites [17].

Encounter rates therefore represent acceptable surrogates for true parrot densities [17], and researchers have already extended their use to estimate the abundance of grey and Timneh parrots in nine countries across their ranges [17,22,23,24,25]. In addition to being simpler and less time-consuming than distance sampling surveys [8,17], encounter-rate methods do not require observers to be trained in accurately estimating detection distances or group sizes. Observers only need to correctly identify the species, making it possible to convert records collected during routine activities of rangers or anti-poaching patrols into encounter rates. This approach creates a link between otherwise anecdotal observations and local abundance estimates in large, difficult-to-access areas that have remained largely unexplored by researchers [17].

One remaining question is whether observations obtained from car transects and point counts can be converted into encounter rates comparable to those proposed by Marsden et al. [17] for walk transects. If so, these approaches could serve as complementary methods, allowing researchers to survey hard-to-reach areas using point counts where long walk transects are impractical, or to cover extensive regions with very low parrot densities using car transects where suitable road networks exist. To test whether encounter rates derived from walk transects, car transects, and point counts yield comparable results, we conducted a nationwide survey of African grey parrots in Equatorial Guinea, combining the three methodologies simultaneously.

## 2. Materials and Methods

### 2.1. Study Area

Equatorial Guinea, officially the Republic of Equatorial Guinea, is located on the Gulf of Guinea on the west coast of Central Africa. The country covers an area of 28,051 km^2^ and comprises a mainland region, the Región Continental (formerly Río Muni), and an insular region that consists mainly of Bioko Island (formerly Fernando Poo) (Figure 1). Equatorial Guinea has a tropical climate characterized by high temperatures, heavy rainfall, and distinct wet and dry seasons. Elevation ranges from sea level to 3008 m a.s.l., and the country spans several ecoregions.

The mainland region lies within the Atlantic Equatorial coastal forests ecoregion, with patches of Central African mangroves along the coast, particularly in the Muni River estuary. Most of Bioko Island falls within the Cross–Sanaga–Bioko coastal forests ecoregion, while the highlands of Bioko belong to the Mount Cameroon and Bioko montane forests ecoregion [26]. In 2024, the country had an estimated population of 1.8 million people, largely concentrated in its two main cities, Malabo (on Bioko Island) and Bata (in the mainland region). Subsistence farming, fishing, and hunting dominate rural livelihoods. Large areas of relatively undisturbed, though predominantly secondary, forest persist throughout the country. In 2018, Equatorial Guinea achieved a mean Forest Landscape Integrity Index score of 7.99 out of 10, ranking 30th globally among 172 countries [27].

### 2.2. Survey Methodology

We conducted fieldwork between 12 January and 23 March 2025 after obtaining the required research permits through a scientific collaboration agreement between our research institution, the Consejo Superior de Investigaciones Científicas (CSIC, Spain), and the Instituto Nacional de Desarrollo Forestal y Manejo de Sistema de Áreas Protegidas (INDEFOR, Government of Equatorial Guinea).

No published information is available on the population status of the African grey parrot in Equatorial Guinea [20]. At the time of expedition planning, only a few records existed in unpublished reports [28] and on citizen science platforms (eBird.org, iNaturalist.org, Observation.org). Consequently, we could not design a stratified sampling scheme based on known distributions or abundances. We therefore adopted a nationwide sampling approach. To implement this design, we established several base camps across the country, covering both the mainland region and Bioko Island. From each base camp, we met with local community leaders to present our permits issued by INDEFOR and to obtain the mandatory authorizations to conduct fieldwork on their lands. When required, we also coordinated with military authorities and logging companies. After securing all necessary permissions, we surveyed the surrounding areas. Eight participants were divided into mixed teams of researchers and INDEFOR technicians to conduct parrot surveys using walk transects, car transects, and point counts, following previously described and widely applied methodologies [7,16,17]. All observers had extensive experience detecting and identifying parrots. We reconfigured team composition daily to carry out car transects, walk transects, and point counts, thereby minimizing potential differences among survey methods attributable to individual observer experience.

For roadside car transects, teams consisting of two observers and a driver operated two four-wheel-drive vehicles at low speeds (20–30 km/h) along unpaved tracks and lightly trafficked paved roads near each base camp. We conducted these transects across all habitat types encountered, excluding strictly urban areas. Teams stopped only when necessary to count parrots. Pairs of observers carried out walk transects along forest trails and along unpaved, low-traffic roads connecting small villages. As with car and walk transects, we did not predefine the location or number of point counts. Instead, we conducted point counts during rest stops at suitable vantage points along transects and during periods when terrain or logistical constraints prevented continued movement.

We conducted all surveys between dawn and 11:00 h and between 15:00 h and dusk, corresponding to periods of peak parrot activity. Adverse weather conditions, including rain, heavy fog, and excessive heat, frequently shortened sampling periods, as such conditions reduce parrot activity and detectability [17]. Accordingly, and consistent with previous studies showing that encounter-rate surveys involve variable observation times [17], survey duration varied substantially because of logistical constraints, trail length, time of day, and weather conditions. We surveyed each transect and point count only once and recorded all surveys, encounters, and individual counts on the citizen science platform Observation (www.observation.org, accessed on 18 November 2025).

Following recommendations for encounter rates derived from walk transects [17], we recorded all parrots detected aurally or visually, whether perched or flying at first detection, for car transects and point counts as well. We calculated encounter rates for each survey method as the number of detections divided by total observation time, expressed as encounters per hour [17]. We also calculated the number of individuals observed per hour as an additional index of relative abundance. For detections based solely on vocalizations, we conservatively assumed the presence of a single individual. For visual or combined visual and aural detections, we estimated straight-line detection distances after training observers with a laser rangefinder (Leica Geovid 10 × 42, Leica, Wetzlar, Germany; range: 10–1500 m).

Before data tabulation and statistical analysis, we grouped all surveys into 10 geographic regions covered during fieldwork. We applied all three survey methodologies simultaneously in eight of these regions: Bioko Island (BK) and seven regions in Río Muni—North West (NW), Central West (CW), Central (C), Central East (CE), South West (SW), South Central (SC), and South East (SE).

### 2.3. Statistical Analysis

We tested the correlation between encounter rates and the number of individuals recorded per hour using Spearman’s rank correlation. We evaluated differences among survey methods in these two abundance metrics and in detection distances using Kruskal–Wallis tests. We assessed differences among survey methods in the proportions of encounters detected aurally or visually, and involving perched or flying birds, using chi-square tests. We used generalized linear models (GLMs) with a normal error distribution and an identity link function to test differences among the three survey methods while controlling for potential confounding factors. We fitted two separate GLMs using log-transformed encounter rates and log-transformed numbers of individuals recorded per hour as response variables. In both models, we included survey method, survey period (morning or afternoon), geographic region where all three survey methods were jointly applied (the eight regions described above), and the interaction between survey method and region as explanatory variables. We conducted all statistical analyses using IBM SPSS Statistics version 27.

## 3. Results

We conducted a total of 199 surveys, including 62 walk transects (320.36 km), 86 car transects (1652.74 km), and 51 point counts, for a total of 192 h of observation (Table 1a). Overall, we recorded 1166 encounters of African grey parrots and a total of 2972 individuals across an elevational range of 0–1385 m a.s.l. (Table 1a).

Most encounters (76.56%) involved parrots detected visually or both visually and aurally. The remaining 23.44% of encounters consisted of parrots detected only aurally, for which group size could not be determined. This proportion did not differ among car surveys (25.35%), walk transects (22.89%), and point counts (21.18%) (χ^2^ = 0.950, df = 2, *p* = 0.621). Among visual and visual–aural detections, encounters with parrots in flight occurred far more frequently (90.87%) than encounters with perched individuals (9.13%). These proportions were nearly identical across car surveys (91.04%), walk transects (90.77%), and point counts (91.18%) (χ^2^ = 0.02, df = 2, *p* = 0.989). The mean estimated detection distance was 82.46 m (range = 9–800 m) and did not differ among car surveys (83.39 m), walk transects (82.14 m), and point counts (83.66 m) (Kruskal–Wallis test, asymptotic *p* = 0.354, N = 795).

The mean encounter rate was 5.03 encounters per hour, ranging from 0 to 52.50 encounters h^−1^ (N = 199). The mean number of individuals recorded per hour was 11.93, also showing substantial variability (range = 0–122.07 individuals h^−1^, N = 199). Encounter rates and the number of individuals recorded per hour were strongly correlated (Spearman’s rank correlation r = 0.96, *p* < 0.001, N = 199).

Surveys conducted in the eight regions where all three survey methods were applied simultaneously (N = 175; Table 1b) showed substantial variability and overlap among results obtained from walk transects, car transects, and point counts (Figure 2). Encounter rates did not differ significantly among the three methods (Kruskal–Wallis test, asymptotic *p* = 0.220, N = 175), nor did the number of individuals recorded per hour (Kruskal–Wallis test, asymptotic *p* = 0.176, N = 175).

A generalized linear model (GLM) fitted to log-transformed encounter rates revealed significant differences among regions (Wald χ^2^ = 59.807, df = 7, *p* < 0.001) and between survey periods (estimate for afternoon surveys = −0.135 ± 0.062 SE; Wald χ^2^ = 4.840, df = 1, *p* = 0.028). Encounter rates did not differ among survey methods (Wald χ^2^ = 1.548, df = 2, *p* = 0.461). The interaction between survey method and region was not statistically significant (Wald χ^2^ = 12.328, df = 14, *p* = 0.580) (Figure 3a).

Similarly, the number of individuals recorded per hour differed significantly among regions (Wald χ^2^ = 54.439, df = 7, *p* < 0.001) but did not differ among survey methods (Wald χ^2^ = 2.060, df = 2, *p* = 0.357) or between survey periods (Wald χ^2^ = 1.958, df = 1, *p* = 0.357). The interaction between survey method and region was also not statistically significant (Wald χ^2^ = 15.330, df = 14, *p* = 0.356) (Figure 3b).

The estimated marginal means of encounter rates and numbers of individuals recorded per hour, derived from the GLMs while controlling for region and survey period, were very similar across survey methods (Table 2) and did not differ significantly in any post hoc pairwise comparison (Wald χ^2^ tests, all *p* > 0.05).

## 4. Discussion

Several methods are available for quantifying parrot population size or abundance. For species that seasonally congregate in a limited number of nesting areas or communal roosts, researchers can estimate total population size by directly counting nests or individuals at roosts [7,29,30,31,32]. However, this approach is not feasible for most parrot species because they are widely and heterogeneously distributed at low densities. For these species, methods that estimate relative abundance are required [7]. Among such approaches, distance sampling surveys have been the most widely used [2], as they allow density estimation by accounting for declining detectability with increasing distance [8]. When surveys are representative and yield unbiased density estimates, researchers can extrapolate them across the species’ distribution to estimate total population size [33,34].

Distance sampling modelling, however, relies on a series of assumptions: transects or points must be positioned randomly with respect to the bird population; observers must detect individuals with certainty; individuals must be detected at their initial locations and must not move naturally or in response to observers before detection; observers must measure distances of detected individuals from the observation point or perpendicular to the transect line accurately; group sizes must be recorded without error; and detections must represent independent events [7,8]. In practice, researchers frequently violate these assumptions, which can lead to biased estimates [21]. Such violations may be particularly common for parrots [9] because they often inhabit dense forest environments where many detections occur aurally, preventing accurate estimation of group size, and because observers frequently detect birds only while they are in flight [7], as shown by our results. In our extensive surveys, nearly one quarter of encounters involved parrots detected exclusively by vocalizations, and 91% of encounters with visual confirmation involved birds first detected in flight. Distance sampling protocols advise against using birds detected in flight [8], yet excluding these detections can substantially underestimate parrot densities [10,15,16].

As demonstrated by Marsden et al. [17], encounter rates not only served as reliable surrogates for African grey parrot densities estimated through distance sampling at the same sites, but also avoided biases arising from violations of distance sampling assumptions. Encounter rates also require less time and fewer resources and can be collected by observers without formal training in distance sampling methods, such as rangers and anti-poaching patrols [17], or even by birdwatchers, provided they can accurately identify and detect the species. Our study further shows that African grey parrot detectability—measured by the relative frequency of aural and visual detections, detections of perched versus flying birds, and detection distances—did not differ among walk transects, car transects, and point counts. More importantly, encounter rates did not differ among the three survey methods when applied jointly in the same areas. Encounter rates also showed a strong correlation with the number of parrots recorded per hour, indicating that both metrics reflect underlying abundance patterns. The number of individuals recorded per hour varied among regions but again did not differ among survey methods. However, encounter rates involve fewer sources of error because purely aural detections do not require estimating group size, and visual estimates of flock size often remain imprecise in forest environments with limited visibility [7].

Our nationwide sampling in Equatorial Guinea captures much of the ecological and population variability reported for the African grey parrot across its global distribution [18,20]. We conducted surveys across a broad altitudinal range and across diverse habitats, including coastal mangroves, lowland rainforests, wetland forests, montane forests, and human-modified landscapes in both mainland and insular regions. As a result of this environmental heterogeneity, the encounter rates we recorded spanned—and in some cases exceeded—the range of values previously reported for this species in other countries [17,22,23]. The relatively high encounter rates likely reflect the country’s limited deforestation and the historically low intensity of parrot capture for international trade (Authors, in prep.).

Another major strength of our study lies in its unplanned design. Following the objective of obtaining reliable encounter rates through casual walks of variable duration [17], we did not determine survey locations or durations a priori. Instead, we adapted them to logistical and environmental constraints encountered in the field. Our approach therefore realistically reflects the challenges faced during parrot survey expeditions in remote regions with limited prior information. Each day, we designed car and walk transects based on the availability of paths and unpaved roads around campsites, while transect duration depended on accessible distances and prevailing weather conditions, such as rain, mist, or extreme heat, which frequently interrupted surveys. We conducted all fieldwork during the dry season; during the rainy season, survey planning would have been even less predictable because of impassable roads, unbridged streams, and frequent rainfall. Despite this lack of formal design, our results demonstrate that all three survey methods produce comparable encounter rates and successfully capture pronounced regional variation. In situations where researchers can implement balanced sampling designs in advance, both in terms of survey number and duration, these methods should yield even more robust results.

The African grey parrot, our focal species, exemplifies a conservation crisis driven by intense capture for trade and widespread forest loss. These pressures have caused numerous local and national extinctions and a global population decline estimated at 50–79% over three generations (43 years), leading to its classification as globally Endangered in 2016 [20]. Despite this status, researchers have made limited progress in quantifying population sizes and trends at regional and global scales [20], largely because of the species’ extensive range and the methodological and logistical challenges of estimating abundance [17]. Population estimates exist for only a few regions, often extrapolated from densities obtained for the now-separated Timneh parrot [20], and no published estimates are available for several countries, including Equatorial Guinea [20]. Callaghan et al. [35] produced a global population estimate of 6,587,487 individuals using eBird data, but the extremely wide confidence interval (95% CI = 69,347–580,214,376) and methodological issues raised by other authors [36] undermine confidence in this estimate. Encounter rates derived from casual walk transects, already applied in several countries [17,22,23], offer a promising avenue for improving estimates of abundance, population size, and long-term trends. Combining walk transects, car transects, and point counts could further accelerate progress in filling these knowledge gaps. In addition, systematic monitoring of sites where parrots congregate from large areas, such as communal roosts [37] and mineral-rich forest clearings [38,39], may help infer long-term population changes.

The three survey methods we tested should not be viewed as alternatives, but as complementary tools for estimating realistic encounter rates. Their combined use provides substantial flexibility for collecting data in difficult-to-sample areas. In some regions, 4×4-accessible tracks are absent; in others, forest trails suitable for walking long distances are scarce; and in many situations, suitable vantage points for point counts are difficult to find. Under such conditions, applying any one of the three methods, either alone or in combination, can yield reliable encounter rates. We recommend reporting encounter rates both for combined datasets and separately for each method to facilitate comparisons with studies that rely on a single approach. An additional implication of our findings is that data collected in previous studies using point counts or road transects can be converted into encounter rates and directly compared with past and future surveys based on walk transects.

Future research should consider applying encounter-rate approaches, ideally combining the three survey methods, to other parrot species that occupy vast ranges and occur at very low densities, making density estimation via distance sampling impractical. For example, the red-fronted parrot (*Poicephalus gulielmi*) is similar in size to the African grey parrot and occurs widely across Central Africa, including the mainland region of Equatorial Guinea [40]. Despite our extensive survey effort, we recorded only three encounters of this species, involving seven individuals in total. Such sparse records nevertheless provide valuable information for species that currently lack any estimates of abundance or population size [40], complicating assessments of their true distribution and conservation status. Researchers should calibrate the relationship between encounter rates and true densities for other parrot species [17] and verify that encounter rates remain comparable across survey methods, as demonstrated here for the African grey parrot. Evaluating the usefulness of encounter rates for other forest species that are readily detected both visually and aurally, such as hornbills [15], also warrants attention.

Finally, future studies should assess whether encounter rates obtained from river-based surveys using small boats or canoes are comparable to those derived from the three terrestrial methods tested here. Many tropical regions are accessible only by river, and boat-based surveys could therefore enable the collection of parrot abundance data in otherwise unsampled areas. Moreover, some species, including several Neotropical parrots [3,18], appear closely associated with waterways, and land-based surveys may underestimate their true abundances.

## 5. Conclusions

In this study, we aimed to test whether walk transects, road transects, and point counts yield comparable encounter rates for estimating the abundance of African grey parrots. Through extensive sampling, we demonstrated that parrot detectability—measured by the proportions of aural and visual detections, the proportions of parrots detected in flight or perched, and detection distances—did not differ among walk surveys, car surveys, and point counts. Encounter rates and the number of individuals observed per hour varied among sampled regions but did not differ among the three survey methods. These results support the use of these three complementary methods for estimating the abundance of this globally endangered species and highlight their potential applicability to many other parrot species.

## Figures and Tables

**Figure 1 biology-15-00073-f001:**
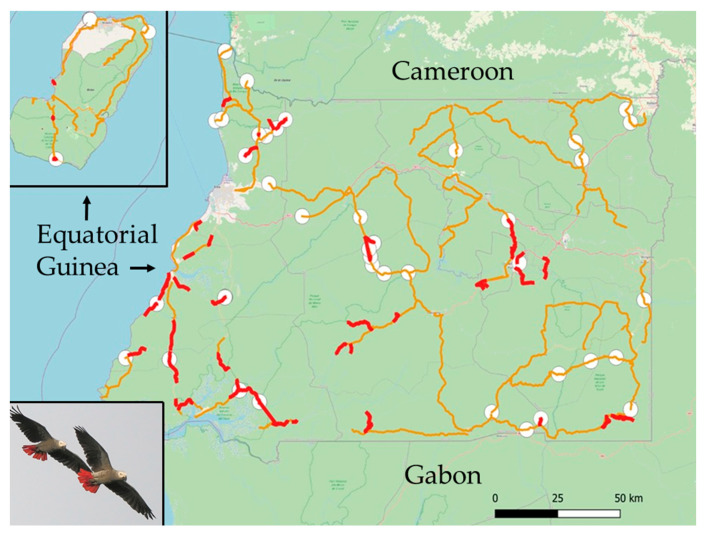
Map of Equatorial Guinea showing the mainland region, surrounded by Cameroon (North) and Gabon (South and East), and Bioko Island (inset). Car transects are shown in orange, walk transects in red, and point counts as white circles. Transects conducted on different days are often combined on the map, which can make them appear as a single continuous transect, and several points and walk transects overlap because of map scale. Photograph: José L. Tella.

**Figure 2 biology-15-00073-f002:**
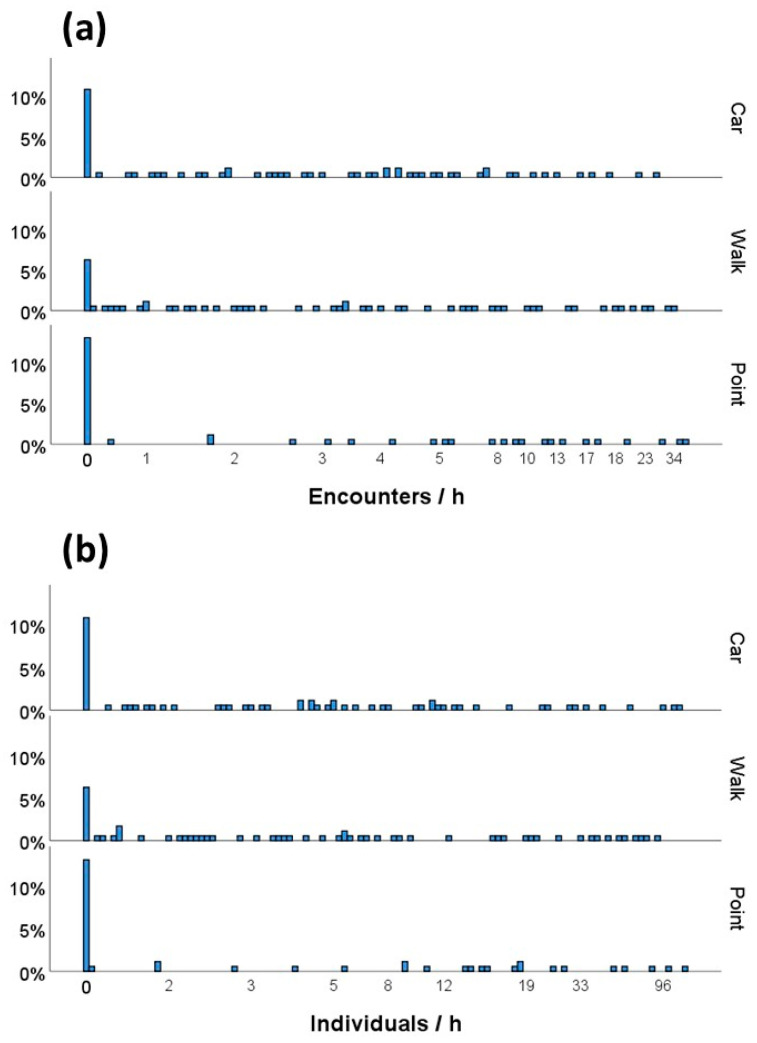
Distribution of raw data for (**a**) encounter rates and (**b**) numbers of African grey parrots recorded per hour across the three survey methods.

**Figure 3 biology-15-00073-f003:**
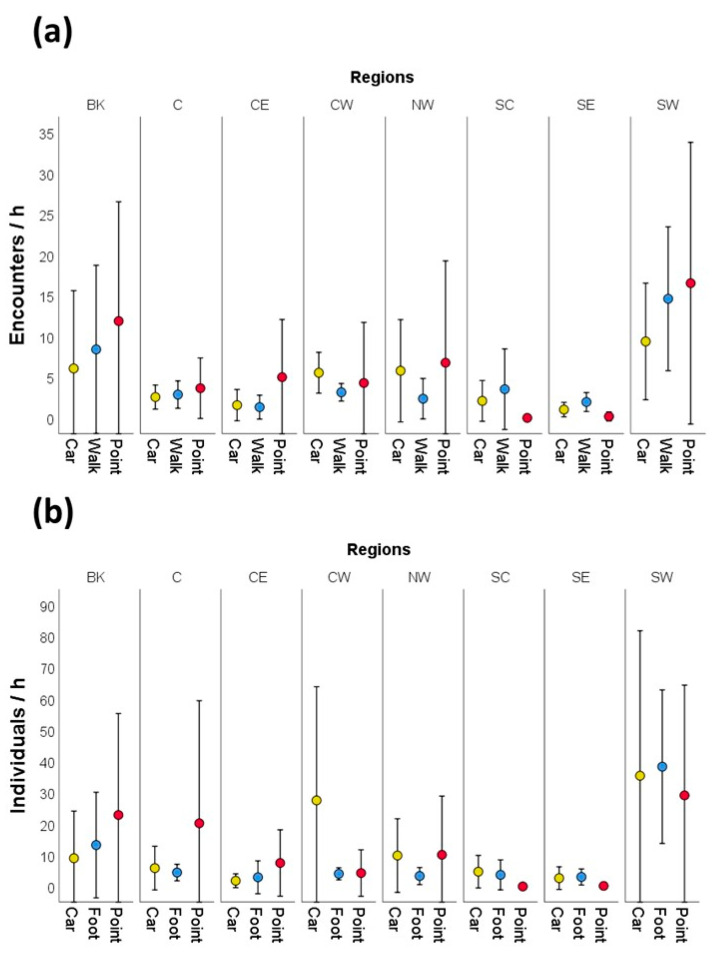
Mean (±SD) encounter rates (**a**) and numbers of African grey parrots observed per hour (**b**) obtained from car transects, walk transects, and point counts across the eight regions where we simultaneously applied the three survey methods (see Section 2 for region abbreviations).

**Table 1 biology-15-00073-t001:** Summary of survey effort and African grey parrot encounters for (**a**) all surveys and (**b**) surveys conducted in the eight regions where all three survey methods were applied simultaneously.

(**a**) **All Surveys**								
**Method**	**N**	**Total Km**	**Mean Km**	**Km/h**	**Total Time (h)**	**Mean Time (h)**	**# Encounters**	**# Individuals**
Walk transects	62	320.36	5.17	3.39	97.68	1.58	785	1739
Car transects	86	1652.74	19.22	22.74	75.68	0.88	296	1026
Point counts	51				18.52	0.36	85	207
Total	199	1973.10			192.00		1166	2972
(**b**) **Surveys Used for Comparing Methods**								
**Method**	**N**	**Total Km**	**Mean Km**	**Km/h**	**Total Time (h)**	**Mean Time (h)**	**# Encounters**	**# Individuals**
Walk transects	62	320.36	5.17	3.39	97.68	1.58	785	1739
Car transects	67	1310.38	19.56	22.74	59.68	0.89	258	747
Point counts	46				16.85	0.37	81	203
Total	175	1630.74			174.21		1124	2689

**Table 2 biology-15-00073-t002:** Estimated marginal means (from GLMs fitted to log-transformed data) for (**a**) encounter rates and (**b**) numbers of African grey parrots recorded using the three survey methods, while controlling for region and survey period.

(**a**) **Encounters/h**			
**Method**	**Mean**	**SE**	**Wald 95% CI**
Walk transects	0.57	0.06	0.46–0.69
Car transects	0.52	0.05	0.43–0.61
Point counts	0.45	0.08	0.31–0.60
(**b**) **Individuals/h**			
**Method**	**Mean**	**SE**	**Wald 95% CI**
Walk transects	0.70	0.07	0.55–0.85
Car transects	0.69	0.06	0.57–0.81
Point counts	0.54	0.10	0.35–0.73

## Data Availability

The datasets presented in this article are not readily available because the data are part of an ongoing study. Requests to access the datasets should be directed to the corresponding author.

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
