# Peer review of "A Comparison of Three Methods for Estimating Abundances of the Globally Endangered African Grey Parrot"

_biology, 2025, doi:10.3390/biology15010073_

Round 1

Reviewer 1 Report

Comments and Suggestions for Authors

Review A comparison of three methods for estimating abundances of the globally endangered African grey parrot

General:

  • Overall, a simple but well executed study with lots of fieldwork (kudos!). Everything is straightforward (e.g., the statistical methods), easy to read, and logical. Maybe my biggest thing is that there could be a but more consideration of what this means, practically, if one wants or needs to combine these methods? It is nice that it is possible, but how would it work if one was to come up with some overall estimates?
    • I would also think that you have to be careful with the word detectability. You looked a specific aspect of it (relative proportion aural/visual) but not at imperfect detection (present but not observed), which would be the main thing I would think you looked at. Maybe add some text to clarify that.
  • This may be a bit of a question out of ignorance on my part, but what would be the benefit of doing point counts (I’m well aware of the method and its benefits by itself) in this specific situation? For example, you mention in 139-140 that ‘the coverage of hard-to-reach areas using point counts where long walk transects are not possible’, but in my own experience: you would still need to walk to each point count location so if it is hard to get to a place on foot (in a Congo basin forest), would you then still not be better off using walk transects and at least gather as much data ‘along the way’? (also clear in Figure 1, all your point count locations are along other transects so what’s the added value?)
  • Some mix of active (“we grouped”) and passive (“was tested”) voice, maybe pick one (active)?

Specific:

13-15: it sounds a bit odd to say it is difficult to apply ‘complex methods’, as this can mean many different things. Maybe just say exactly which methods are difficult to apply (or at least a ‘such as’)?

99-104: you write ‘on the other hand’ as if you present a contrast (first an ‘on the one hand’ followed by an ‘on the other hand’) but that’s not the case. ‘In addition’ would seem more appropriate? But then, what are these assumptions you speak of (in line 103)?

344-346: elsewhere you also refer to detectability did not differ between the methods (e.g., 439) but only here is it clear that you’re talking about the relative proportions of aural and visual detections. However, detectability in terms of how many were missed that were there (imperfect detectability a la Monitoring programs need to take into account imperfect species detectability - ScienceDirect) could still be very well present, right?

Comments on the Quality of English Language

Maybe choose a voice (active vs passive) and check the paper one more time for oddities such as "The interaction between methods and regions neither was statistically significant" (lines 288 and 292).  

Also, be very concrete and precise and avoid things like 'complex methods', just say exactly which ones. 

Reviewer 2 Report

Comments and Suggestions for Authors

The authors present an interesting study on estimating abundances of the globally endangered African grey parrot. Different estimation methods used to understand which can be used for the future studies. In this context, the study contributes to understanding the abundance of African grey parrot and also which estimation method would be used. However, some parts of the manuscript need to be improved. Therefore, I would like to make some suggestions to improve the quality of the paper as below:

Lines 12-27: The Simple Summary Section should be shortened. The main aim of the study, the method, the most important findings, and the main conclusion can be given with 5-7 sentences.

Lines 62-64: “However, obtaining sufficient and reliable population estimates remains a significant challenge, and parrots (order Psittaciformes) pose one of the most striking cases in this regard.” A reference is needed

Lines 64-68: “While they are one of the most diverse groups of birds, with around 400 species distributed throughout tropical and subtropical regions of the world, they are also the group with the highest percentage of globally threatened species (Olah et al. 2016, McClure and Rolek 2020), accounting for around 30% of the species (IUCN 2025).” The sentence is too long. Please re-write here with shorter separated sentences.

Lines 90-91: “the alarming population declines driven by threats such as habitat loss, capture for the pet trade, or persecution due to crop damage”. Illegal trading should also be emphasised here.

Lines 116-117: “Trying to get ahead of these problems, which are inherent to many other parrot species around the world”. Please rephrase this sentence.

Lines: 136-145: This part of the paper is important since the authors should explain the purpose of the study and their hypothesis (I mean; what is the problem and what did you do to solve this problem) are given here. In this context, please explain clearly which gaps intent to filled by the authors with 1-2 sentences. In this way, the bridge between the problem and the study aim would be stronger. Before this paragraph, habitat choice and basic biology of the African grey parrot can be mentioned.

Lines 144-145: “we found that encounter rates did not differ among the three methodologies used, suggesting that they can be used as complementary methods to obtain parrot abundances”. This sentence is a result so it should be better in results and/or conclusion sections.

Lines 200-202: “Walk transects were also conducted by two observers, traversing trails that crossed forests, as well as unpaved and low transit roads connecting small villages.”. Please explain, if car and walk transect performed in same or same person performed both surveys in different time.

Lines 232-246: Please add proper references for the 2.2. Statistical analysis subsection.

Lines 239 -241: “Generalized Linear Models (GLMs), using normal distribution and identity function, were used for testing differences between the three survey methods while controlling for other potential, confounding effects.” A reference is needed.

Lines 306-337: The discussion section can be enriched with a more theoretical interpretation and relate the present results with additional concepts. For instance, the study results can be discussed with similar studies from different methods and animal species in the broader context. Also, the different counting methods such as satellite imaginary, remote devices, camera-traps or drones etc. should be discussed in this section.

Lines 438-444: The conclusion section should be rephrased as follows; please start with a brief description of the study (the aim of the study with a sentence), explain the main findings of the study briefly (the results that the authors found), explain how your results contribute to field with 2-3 sentences, and explain the limitations of the study and describe the future remarks briefly.

Comments on the Quality of English Language

Some parts of the manuscript are not easy to understand (mentioned below in specific comments). There are many long sentences and wordiness. This situation disrupts the flow of the subject and the continuity of the reading. Because of this reason, authors should reconsider writing some parts of the manuscript.

Round 2

Reviewer 2 Report

Comments and Suggestions for Authors

The authors improved the manuscript with the mentioned comments.